# Patterns of Diversity, Structure and Local Ecology of Arthropod-Pathogenic Fungi in the Amazonian Forest of Cusco and Madre de Dios Regions, Southern Peru

Isau Huamantupa-Chuquimaco [1,2,3,*], María Encarnación Holgado Rojas [2], Miguel Angel Quispe Ordoñez [2], Mishari García Roca [4], Anatoly Cárdenas Medina [5], Willians Quispe Ancco [2], Roger Oswaldo Poccohuanca-Aguilar [3], Zoila Magaly Cuba Córdova [6,7], Jackeline Greta Meza Calvo [2] and Tatiana Ibeth Sanjuan Giraldo [8]

1   Herbario Alwyn Gentry (HAG), Universidad Nacional Amazónica de Madre de Dios (UNAMAD), AV. Jorge Chávez N°1160, Puerto Maldonado 17001, Madre de Dios, Peru
2   Laboratorio de Micología Aplicada CIPHAM, Universidad Nacional de San Antonio Abad del Cusco, Av. La Cultura, Cusco 773, Cusco, Peru; encarnacion.holgador@unsaac.edu.pe (M.E.H.R.); qorikamayoq@gmail.com (M.A.Q.O.); 114215@unsaac.edu.pe (W.Q.A.); jackelinemzc@gmail.com (J.G.M.C.)
3   Departamento de Ciencias Básicas, Universidad Nacional Amazónica de Madre de Dios (UNAMAD), AV. Jorge Chávez N°1160, Puerto Maldonado 17001, Madre de Dios, Peru; rpoccohuanca@unamad.edu.pe
4   Departamento Académico de Ingeniería Forestal y Medio Ambiente, Universidad Nacional Amazónica de Madre de Dios (UNAMAD), AV. Jorge Chávez N°1160, Puerto Maldonado 17001, Madre de Dios, Peru; mgarcia@unamad.edu.pe
5   Colección Científica de Ictiología, Universidad Nacional Amazónica de Madre de Dios (UNAMAD), AV. Jorge Chávez N°1160, Puerto Maldonado 17001, Madre de Dios, Peru; tolyskrdenas@gmail.com
6   Departamento Académico de Biología, Universidad Nacional de San Luis Gonzaga de Ica, Av. Los Maestros S/N Ciudad Universitaria Km, Ciudad Ica 305, Ica, Peru; commicarpus@gmail.com
7   Dirección de Humanidades, Campus Ica, Universidad Tecnológica del Peru (UTP), Av. Ayabaca S/N, Sector San José, Ciudad Ica 305, Ica, Peru; C20391@utp.edu.pe
8   NPO Grupo Micólogos Colombia, Calle 116A # 71A-32, Bogotá 111121, Colombia; contacto@grupomicologoscolombia.org
*   Correspondence: ihuamantupac@unamad.edu.pe

**Abstract:** The ecological role and potential management of arthropod-pathogenic fungi (APF) in neotropical forests are of great importance, but they are still little studied. The present study achieves a first estimation of diversity patterns, structure and local ecology of APF in the Amazonian forests of the Cusco and Madre de Dios regions in southern Peru. We sampled 39 localities in five basins, examining 277 specimens, four families and 20 genera with 82 species (40% morphospecies). The most diverse families were Cordycipitaceae with 51 species and Ophiocordicipitaceae (22). Cusco obtained a greater diversity: four families, 18 genera and 58 morphospecies, with the Urubamba and Amarumayu basins having greater diversity (31 and 20 species); for the Madre de Dios basin, there was 28 species. In both regions, the richness values were corroborated by Fisher's Alpha and Chao-1 indexes, the latter identifies Amarumayu and Araza with maximum values. The NMDS analysis showed a good pattern of separation of the two APF communities, although an important group was shared. Elevation was identified as the environmental variable with the strongest influence on diversity and structure. The dominance analysis identified *Ophiocordyceps australis* and *Paraisaria amazonica* as hyperdominant, due to their density and distribution. The local ecological patterns in Pongo de Qoñec show that the richness of entomopathogens is largely favored by low understory light, associated with pristine or little-impacted habitats. We conclude that this first approximation of the knowledge of the high diversity of APF in southern Peru is still insufficient, but it demonstrates the importance of their conservation and represents enormous potential for sustainable management.

**Keywords:** *Cordyceps*; *Ophiocordyceps*; piedmont forest; terra firme forest

## 1. Introduction

The fungi kingdom is a significant and varied group of eukaryotic organisms in both terrestrial and aquatic ecosystems. Fungi, being heterotrophic organisms, are incapable of generating their own food and are therefore forced to establish different ecological interactions with other organisms, such as saprotrophy, mutualism and parasitism [1]. Within the saprotrophic and parasitic interactions are arthropod-pathogenic fungi (APF), organisms with the particular ability to infect arthropods, such as insects, spiders and some mites, whose health status was initially healthy, and then cause disease, eventually leading to their death [2]. These fungi belong to the phyla Entomophthoromycota, Basidiomycota and Ascomycota. The latter phylum comprises over 750 species, which parasitize a range of arthropods such as beetles, flies, butterflies and spiders [3]. Some species of APF are currently sources of biochemical substances with interesting biological and pharmacological properties such as cordycepin and cyclosporine, B group vitamins (B1, B2 and B12), vitamin E and vitamin K and numerous minerals such as sodium, zinc, iron, potassium, calcium, manganese and selenium [4,5].

In the ascomycetes, the order Hypocreales contains the major diversity of APF, which are distributed in three families, Cordycipitaceae, Clavicipitaceae and Ophiocordycipitaceae, and atomized into 124 genera [6,7]. In general, the greatest diversity of APF is found in environments with little human disturbance, such as primary forests, associated with particular environmental variables [8,9]. Although other authors mention that a high diversity can be recorded even in impacted areas in tropical forests, as occurs with the genus *Cordyceps* on ant hosts [10]. The diversity of APF fungi is still little studied in southern Peru, including reports by [11] and [12] who evaluated several localities in the Madre de Dios, Río Los Amigos and Tambopata basins in the Madre de Dios region, reporting *Cordyceps* s.l. as the most diverse entomopathogenic fungi family, with the species *Paraisaria amazonica* and *Ophiocordyceps australis* among the most frequent and present in various habitats.

Therefore, the objective of this research was to contribute to the knowledge of the diversity patterns, composition, abundance and dominance of arthropod-pathogenic fungi and their relationships with environmental and biological variables at the regional and local level in the Amazonian forests of Cusco and Madre de Dios regions.

## 2. Materials and Methods

### 2.1. Study Area

Our study area comprises the Amazonian forests of the regions of Cusco and Madre de Dios, located in southern Peru (Figure 1). In Cusco, arthropod-pathogenic fungi (APF) were evaluated in 16 localities (Tables 1 and 2) located in the basins of Alto Madre de Dios (Amarumayu) in the province of Paucartambo, district of Kosñipata, the Araza basin in the province of Quispicanchis, district of Camanti, the Urubamba basin in the province of La Convención and district of Echarati. These areas are characterized by an altitudinal gradient ranging from 380 masl to 1100 masl, recognized as Amazonian foothill forests of the southern Peruvian Yungas or the pre-montane Amazonian rainforest (Amazonian piedmont forest), with an average annual rainfall of 3200 mm [13]. For the Madre de Dios region, 23 localities were included (Tables 1 and 2) in the basins of Inambari in the province of Tambopata, Mazuco district, the Madre de Dios basin in the province of Tambopata, the districts of Las Piedras and Tambopata, and the Tambopata basin in the province and district of the same name; in this region, the forests are characterized as being flat with altitude ranges from 230 to 450 masl, and the forest type corresponds to terra firme, with an average temperature of 24 °C. The average annual rainfall is 1847.1 mm [14].

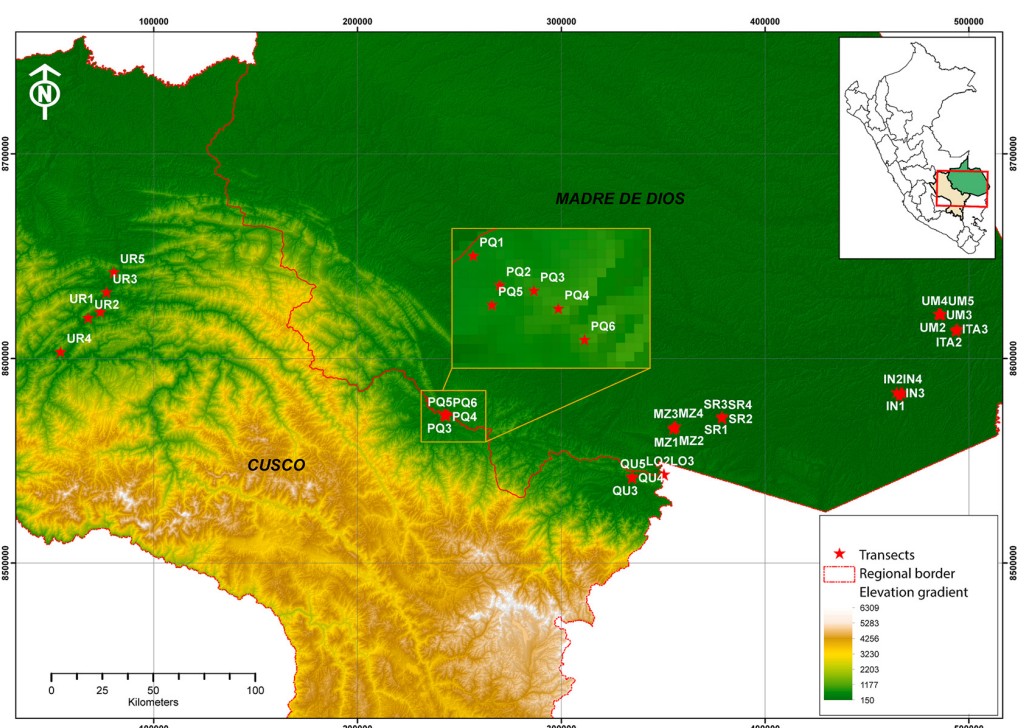

**Figure 1.** Map showing the location of the sampling areas in the watersheds of the localities in the regions of Cusco and Madre de Dios.

**Table 1.** Biogeographical and environmental data of the entomopathogens evaluated in the Amazon regions of Cusco and Madre de Dios province.

| Region | Basin | Type of Amazonian Forest | UTM | East Coordinates | North Coordinates | Elevation (msl) | Annual Total Precipitation (mm) | Average Annual Minimum Temperature (°C) | Average Annual Maximal Temperature (°C) |
|---|---|---|---|---|---|---|---|---|---|
| Cusco | Araza | Piedmont | 19L | 334894.07 | 8542052.28 | 600.00 | 4670.00 | 16.00 | 28.83 |
| Madre de Dios | Inambari | Terra firme | 19L | 350424.55 | 8543401.24 | 400.00 | 2600.00 | 19.33 | 28.92 |
| Madre de Dios | Tambopata | Terra firme | 19L | 467402.00 | 8583322.00 | 263.00 | 2160.00 | 20.58 | 30.67 |
| Cusco | Urubamba | Piedmont | 18L | 706345.52 | 8606087.83 | 750.00 | 3200.00 | 21.33 | 31.25 |
| Cusco | Amarumayu | Piedmont | 19L | 242571.88 | 8572773.15 | 650.00 | 3950.00 | 15.00 | 29.17 |
| Madre de Dios | MadreDios | Terra firme | 19L | 486657.46 | 8621633.58 | 250.00 | 1900.00 | 20.42 | 30.83 |

**Table 2.** Quantitative data on entomopathogen diversity in the watersheds of the Cusco and Madre de Dios regions.

| Basin | Type of Amazonian Forest | Number of Samples Evaluated | Species Richness | Fisher Alpha Diversity | Chao-1 Diversity |
|---|---|---|---|---|---|
| Amarumayu | Piedmont | 6 | 20 | 11.9 | 71.49 |
| Araza | Piedmont | 5 | 16 | 11.8 | 37.84 |
| Inambari | Terra firme | 11 | 15 | 7.162 | 20.15 |
| Madre de Dios | Terra firme | 8 | 28 | 13.29 | 50.51 |
| Tambopata | Terra firme | 4 | 14 | 5.58 | 21.38 |
| Urubamba | Piedmont | 5 | 31 | 17.76 | 35.53 |

Habitats in the Cusco region were more variable, associated with a wide Amazonian altitudinal gradient, where rainfall and humidity regimes are also more variable than those

terra firma forests from Madre de Dios, where they are more homogeneous with a marked seasonality of dried and rainy months (Table 1).

## 2.2. Methodology

Twenty-one field samplings were carried out in the Amazonian forests of the Cusco and Madre de Dios regions; the samplings were carried out in different periods from 2019 to 2023 year. Each sampling was developed in different localities of the mentioned basins for each region (Table 1, Figure 1). Collections were made following the standardized protocols for AFP [9,10] and consisted mainly of transects of approximately 2 km in each locality. In each transect, an intensive search for AFP was carried out, with data collection of the following variables: host, number of individuals, substrate where the host was found, description of the sexual or asexual stage (teleomorphic or anamorphic state), morphometric measurements, type of habitat and, as environmental variables, elevation and coordinates. To know the patterns of local ecology we have worked with the records made in the locality of Pongo de Qoñec (Amarumayu basin), where variables of light intensity and type of forest were evaluated attributing to the degree of impact.

The specimens were collected and carried to the CIPHAM laboratories at the Universidad Nacional de San Antonio Abad del Cusco for identification, isolation and storage. The identifications were based on the morphological characterization of the stroma, asci, ascospores and part-spores for each specimen in the case of the ascomicetes. Isolations were made from a piece of stroma hanging from the lid of a Petri dish with tap water media (AW). In the case of the asexual form, a print of the synnema was placed into the AW media. The purifications were made in potato dextrose agar (PDA), Sabouraud agar (SDA) and malt extract agar (MEA) for 7 to 14 days or until the conidiogenic state was observed. These data as well the growth and development of mycelia characteristics were used to complete the morphological identification. Species classification follows a specialized bibliography for APF such as [15,16]. Taxonomic standardization was based on the Index Fungorum (https://www.indexfungorum.org/names/names.asp, accessed on 12 November 2022). The data of mean annual precipitation and mean annual maximum temperature were obtained using the extract points function, through each coordinate with the Qgis software (2020). The database came from the Worldclim layers of the year 2021.

## 2.3. Data Analysis

Alpha diversity was calculated using quantitative data from the total values at each transect of the basin localities, which were used to calculate Fisher's alpha index ($\alpha$). Fisher's alpha index is an abundance model derived from a logarithmic series using only the number of species (S) and the total number of individuals (N) [17]. This index is expressed as $S = \alpha \ln [1 + (N/\alpha)]$, where S = number of species in the sample and N = number of individuals in the sample. Furthermore, to estimate the total number of species that can be found in each region, abundance curves were developed using the rarefaction method, with the extrapolation function at the level of each region. For this part of the analysis, Hill numbers q = 0 were used to quantify the effective number of all species, including rare species and q = 1 (Shannon exponential diversity) as the effective number of common entomopathogens [18]. These analyses were run using the iNEXT package developed for R software version 4.2.1 [19]. To complement the estimation of diversity in each catchment, we applied the non-parametric Chao-1 estimator. This estimator prioritizes the presence of rare species within a sample (in this case, each catchment locality), with the formula $Chao1 = Sobs + ((n - 1/n) F1(F1 - 1)/2(F2 + 1))$, where Sobs is the number of species observed in each locality, n the number of samples, F1 is the number of species observed in a single locality (singleton species) and F2 is the number of species observed in two localities (doubleton species) [20,21]. In order to know the beta diversity and structure, the non-metric multidimensional scaling analysis (NMDS) was applied, with two axes of expression, considering the Bray–Curtis dissimilarity with the data of abundances and environmental variables. According to the results, the stress was considered, with

permutation tests (N = 999) and the percentage of ordination contributions in the 2 axes. To obtain the NMDS results, we used the envfit and metaMDS functions implemented in the vegan package [22], in the statistical software R programming environment [19].

To structure the abundance and dominance patterns, we generated rank abundance curves [23], where we observed those with the highest proportion and frequency. This analysis was developed in the statistical software PAST version 4.09 [24]. Similarly, to estimate the dominance, the criteria proposed by [25] were adapted, where a hyperdominant is one that is found with more than one individual in all the sites sampled (in our case, in the localities of each basin). To complement these patterns, we analyzed the differences or similarities existing in both ecosystems, using the non-parametric Mann–Whitney U test to determine if there were differences in species richness and abundance between localities. These analyses were carried out in the R programming environment [19].

To determine the patterns of relationships between biological and environmental variables (luminosity, precipitation, elevation and temperature), linear regression analyses were performed. These analyses and graphs were developed in the statistical software R programming environment [19].

## 3. Results

### 3.1. Diversity and Composition

For the Amazonian forests sampled in the regions of Cusco and Madre de Dios, 277 specimens of arthropod-pathogenic fungi were collected. They belong to three families from the Ascomycota phylum, Clavicipitaceae, Cordycipitaceae and Ophiocordycipitaceae, with 19 genera and 82 species (40% including morphospecies). The phylum Entomophthoromycota was represented by the *Massospora* genus. The families with the largest species diversity were Cordycipitaceae with 51 species and Ophiocordycipitaceae with 22. The most representative genera were *Cordyceps* with 20 species, *Ophiocordyceps* with 16, and *Akanthomyces* and *Beauveria* with 8 species each. At the species level, the most abundant were *Ophiocordyceps australis*, *Paraisaria amazonica*, *Ophiocordyceps curculionidae*, *Beauveria* sp3, *Gibellula* sp1, *Beauveria acridophila* and *Cordyceps tenuipes* (Figure 2) (Table S1).

According to the evaluated localities in the six basins, four families were identified in the Cusco region (Clavicipitaceae, Cordycipitaceae, Ophiocordycipitaceae and Entomophthoraceae), with 18 genera and 58 species (24.6% of morphospecies), and *Cordyceps* with 11 species and *Ophiocordyceps* with 10 were the most diverse. The most abundant species were *Ophiocordyceps australis*, *Paraisaria amazonica*, *Ophiocordyceps curculionidae*, *Beauveria* sp3, *Gibellula* sp1, *Samsoniella* sp1, *Massospora* sp1 and *Paraisaria gracilis*. For the Madre de Dios region, the families Clavicipitaceae, Cordycipitaceae and Ophiocordycipitaceae were identified; 10 genera were recorded, and the most diverse were *Cordyceps* and *Ophiocordyceps* with 11, respectively; and 39 species were recorded, of which the most abundant were *Ophiocordyceps australis*, *Paraisaria amazonica*, *Beauveria acridophila*, *Ophiocordyceps curculionidae*, *Beauveria locustiphila* (Figure 2B) and *Cordyceps* sp3.

The most species-rich basins were Urubamba with 31, Madre de Dios with 28 and Amarumayu with 20 species. According to the Fisher Alpha diversity index, the most diverse basins were Urubamba with 17.76, Madre de Dios (13.29) and Amarumayu with 11.9 (Table 2). According to the Chao-1 index, which projects the diversity from abundance and rarity, the Amarumayu basin can host up to 71 species, Madre de Dios up to 50 and the Araza up to 38 species; the latter Chao-1 weights for the presence of species with few individuals (Table 2). Considering species richness, the Araza basins with 16 species, Inambari (15) and Tambopata with 14 can be considered as having regular richness. This same pattern is confirmed by Fisher's Alpha index and to a large extent by Chao-1 (Table 2).

At the regional level, considering that both the basins and localities of the Cusco region comprise Amazonian piedmont forests and those of Madre de Dios comprise terra firme lowland, the rarefaction curves using the Hill index revealed for q = 0 a projected diversity of up to 90 species for Cusco and up to 56 species for Madre de Dios (Figure 3A), and

using the Hill index q = 1 considering the most common species, for Cusco projects up to 65 species and for Madre de Dios up to 45 species (Figure 3B).

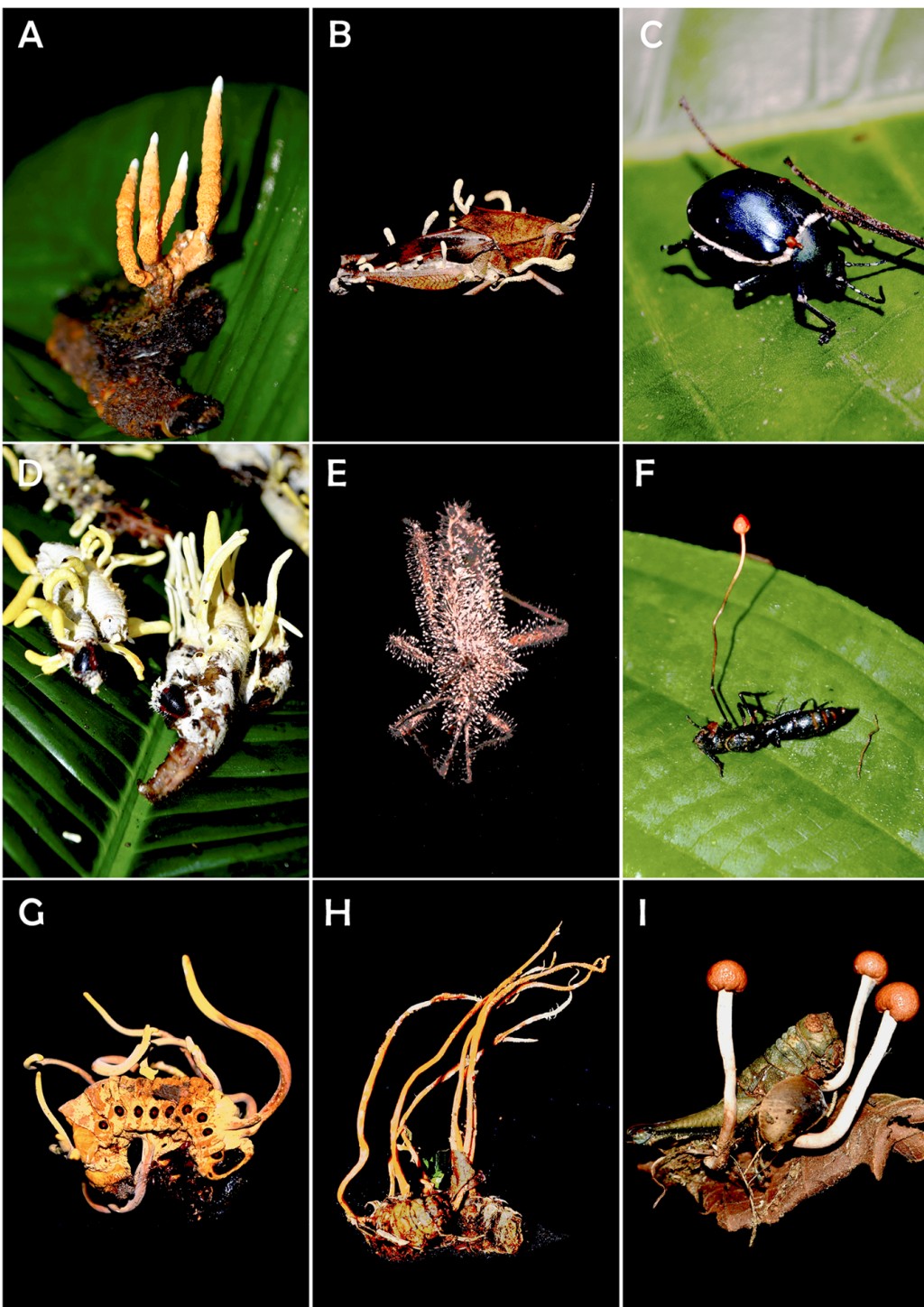

**Figure 2.** Some frequently evaluated entomopathogenic species. (**A**) *Nigelia martialis*, (**B**) *Beauveria locustiphila*, (**C**) *Beauveria* sp1, (**D**) *Cordyceps tenuipes*, (**E**) *Isaria* sp5, (**F**) *Ophiocordyceps australis*, (**G**) *Ophiocordyceps melolonthae*, (**H**) *Ophiocordyceps* sp, (**I**) *Paraisaria amazonica*.

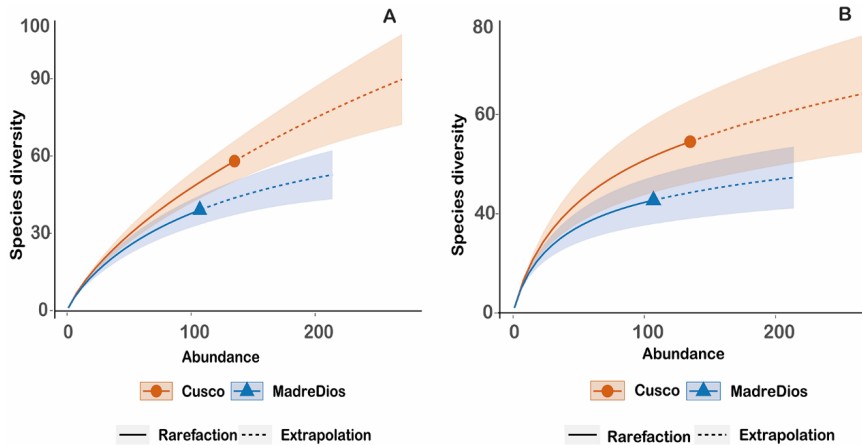

**Figure 3.** Accumulation curves for the two Amazonian regions, solid lines indicate the rarefaction method, with evaluated species and the extrapolation method (dashed lines), shaded areas correspond to the 95% confidence bands. (**A**) Accumulation curve considering the rare species (Hill q = 0) and (**B**) accumulation curve considering common species (Hill q = 1).

### 3.2. Structure

Regarding the structure of the APF populations, the NMDS analysis (n = 82) shows a stress value of 0.07, which indicates that the result is a good representation of the distances in the original distance matrix, and therefore, there is clear evidence of two separate groups considering the composition, abundance and environmental variables. The environmental variables are represented by the vectors and the strength of these show the distance and direction of the correlation of each character in relation to the NMDS axes. Of these, the one that contributes most to the conformation of the groups is the elevation (altitude), with more than 72%, which associates several taxa to this in the forests of the Amazonian foothills (Figure 4).

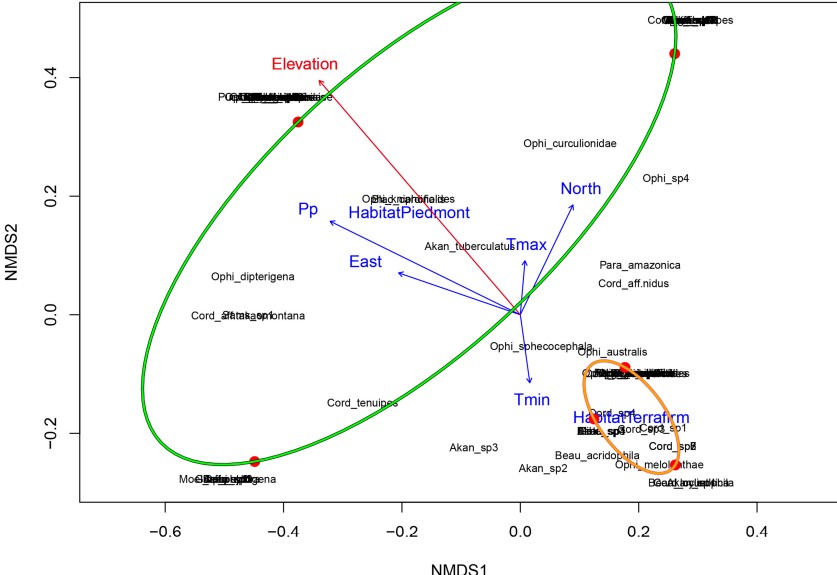

**Figure 4.** NMDS ordination pattern for the assessed localities in the watersheds, Amazon foothills (Amarumayu, Araza, Urubamba) grouped in the green ellipse and watersheds for lowland Amazon (Inambari, Madre de Dios and Tambopata) grouped in the orange ellipse. Stress = 0.07; ellipses are the 95% confidence interval around the centroid of each group. Environmental variables are represented by blue and red vectors (Pp, East and North coordinates, Tmx = average monthly maximum temperature, Tmin = average monthly minimum temperature and elevation).

The range abundance curves (Figure 5A,B) show that for the Amazonian forests of the Cusco region, the species with the highest abundances were *Paraisaria amazonica* with 18 individuals contributing 10.2% to the total, *Ophiocordyceps australis* (15 ind., 8.5%), *Gibellula* sp1 (10 ind., 7.3%), *Ophiocordyceps curculionidae* (9 ind., 6.4%) and *Cordyceps tenuipes* (5 ind., 5.7%), which together account for more than 35% of the total abundance. For the Madre de Dios region, the most representative species were *Ophiocordyceps australis* with 22 individuals and 12.3% of the total, *Paraisaria amazonica* (14 ind., 9.4%), *Beauveria acridophila* (8 ind., 7.8%), and *Ophiocordyceps curculionidae*, *Beauveria locustiphila* and *Cordyceps* sp3 each with 4 individuals and 5.5% of the total. Together they represent more than 45% of the total abundance.

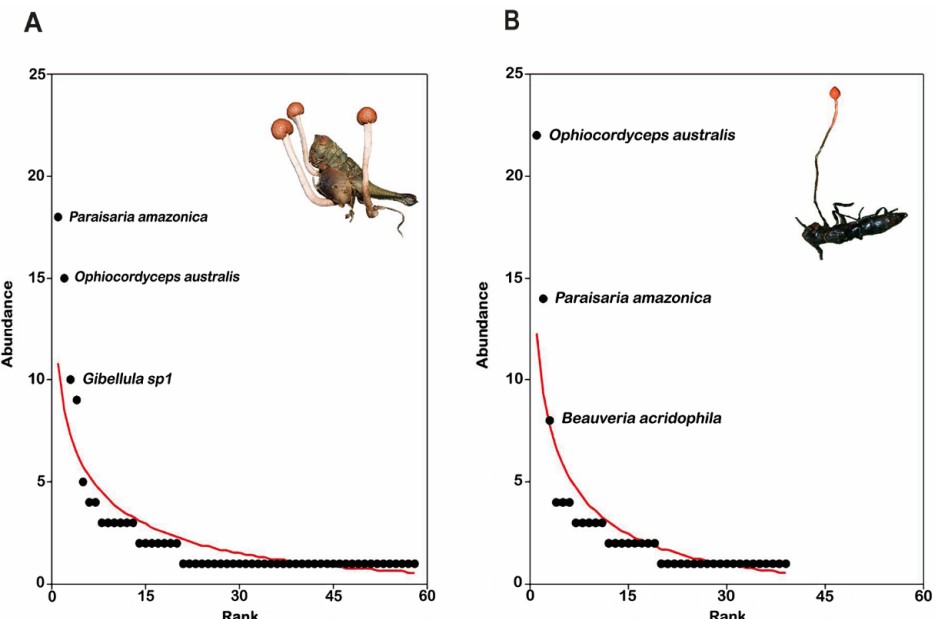

**Figure 5.** (**A**). Range abundance curve for the Cusco region and (**B**) range abundance curve for the Madre de Dios region.

### 3.3. Environmental Relationships

Linear regression analysis showed that altitude (elevation) is the environmental variable that best explains the relationship between altitudinal gradient and species richness (R = 0.31, $p$ = 0.057) (Figure 6A), while the other variables showed a low relationship between precipitation and species richness (R = 0.23, $p$ = 0.16) (Figure 6B), and even lower were the relationships between the maximum average annual temperature with species richness (R = 0.16, $p$ = 0.32) (Figure 6C) and finally minimum average temperature with species richness (R = 0.017, $p$ = 0.92) (Figure 6D).

At the local level in the Pongo de Qoñec locality (Amarumayu basin), luminosity (light input to the understory in each evaluated sample) shows a good negative relationship (R = −0. 76, $p$ = 0.0001) (Figure 6E); this is explained by the fact that entomopathogen species richness is higher in primary forest habitats. In secondary forests with recent and old (10–12 years) anthropogenic impacts, the species richness decreases, but a few species are represented by a considerable abundance, as observed for *Ophiocordyceps australis* (Figure 6F).

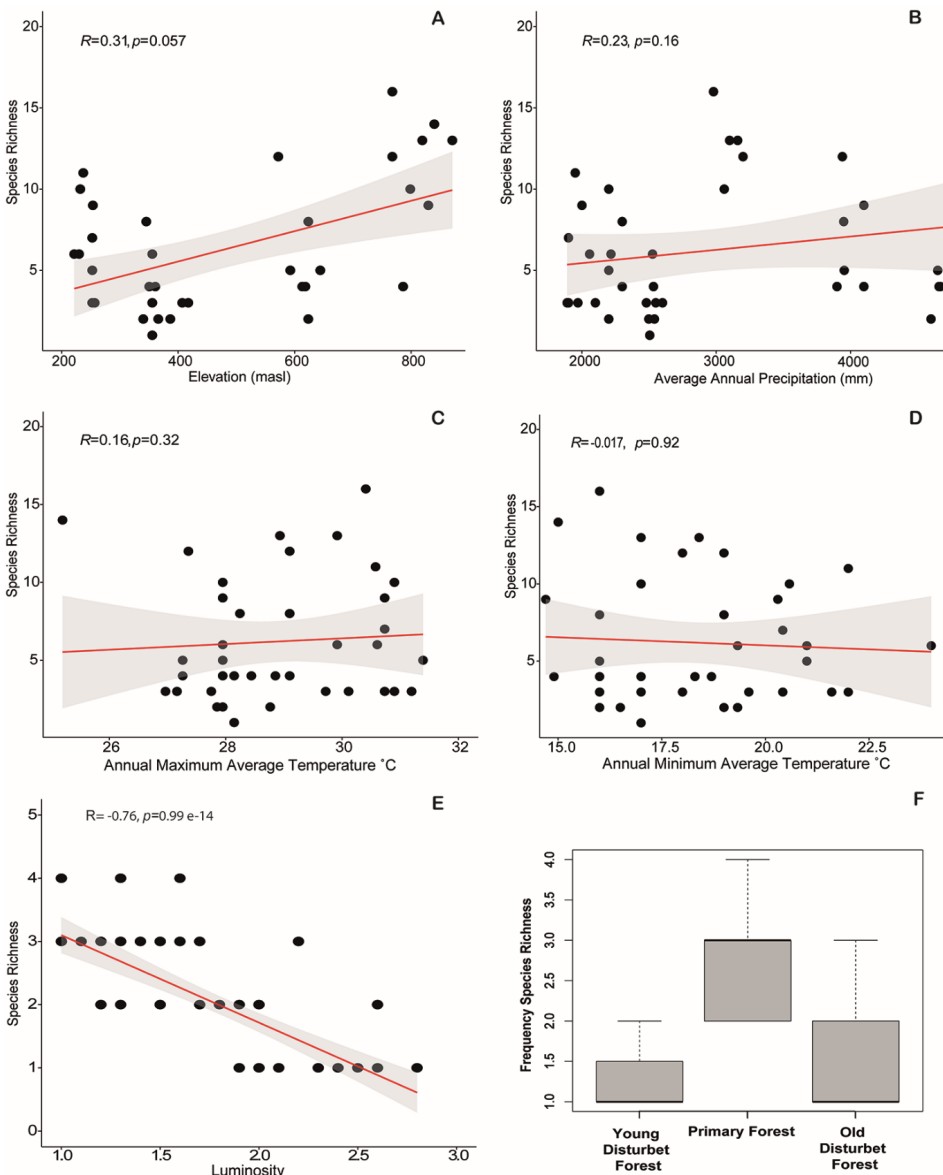

**Figure 6.** Linear regression and box plots. (**A**). Elevation and species richness, (**B**) average annual precipitation and richness, (**C**) average annual maximum temperature and richness, (**D**) average annual minimum temperature and richness, (**E**) luminosity and richness and (**F**) box plots of the frequency of species present in the forest types.

## 4. Discussion

### 4.1. Patterns of Diversity and Composition

The study identified 82 morphospecies of arthropod-pathogen fungi (APF) in Southern Peru, particularly in the Amazonian forests of the Cusco and Madre de Dios regions. This report presents a significant finding. Although this initial estimation of species diversity remains inconclusive, consisting of approximately 40% morphospecies, it is possible that several unidentified species may be included. These species are broadly distributed in the order Hypocreales, having species in three families of the eleven families in this taxonomic rank, Ophiocordycipitaceae, Cordycipitaceae and Clavicipitaceae, which are considered the most diverse families of APF in tropical forests [26]. At the genus level, *Cordyceps* s.s. with 20 species and *Ophiocordyceps* with 16 are the most representative, as well as *Akanthomyces* and *Beauveria*, which in other works are also referred to as the most diverse in neotropical forests, such as in the Colombian and Ecuadorian Amazon [26,27]. *Akanthomyces* with eight morphospecies has a high richness and is distributed in the lower Amazon foothills.

The variation in morphology exhibited by specimens of this genus, as well as the challenges in their full identification, suggests the presence of species complexes within this fungal type. Such complexes can solely be differentiated by means of molecular phylogenetic analysis, as is the case in the *Ophiocordyceps* genus [28]. Remarkably, the genus richness values of *Ophiocordyceps* and *Cordyceps* s.s. in this case appear to exceed those of some highly diverse tropical regions, such as the Amazon Basin from Ecuador, where up to 15 morphospecies of *Ophiocordyceps* and 15 species of *Cordyceps* s. l. were identified in deciduous woods from Mexico [27,29].

The Amazonian foothill forests of Cusco exhibit a higher diversity with 58 identified species compared to 39 in Madre de Dios. Both regions have high levels of endemism and richness of taxa that converge from montane and Amazonian forests. In reference to this condition, for example to the plants, the Cusco region is identified as the third richest and most endemic region in Perú, especially when taking into account the Amazonian foothill forests in the provinces of La Convención, Paucartambo and Quispicanchis [30]. The pattern appears to be consistent for arthropod-pathogenic fungi, as these are the regions where the Alpha Fisher index is highest and the Chao-1 index is expected to be present in over 50% of the reported species (up to 90 species).

The density and frequency of APF species is greater in the forests of Madre de Dios when compared to those of Cusco, especially in the Terra firme forest. This could potentially be attributed to the confluence of the basins of different rivers in the mentioned forest. Species such as *O. australis*, *P. amazonica*, *O. curculionidae*, *B. acridophila*, *O. sphecocephala*, *C. tenuipes*, *B. locustiphila* and *O. melolonthae*, and morphospecies such as *Akanthomyces* sp2 and *Cordyceps* sp3 stand out among them, and they were reported previously in Los Amigos basin [11].

### 4.2. Dominance Patterns at the Regional Scale

In terms of dominance, the obtained data are very revealing and interesting, due to there being only two hyperdominant species in this Amazonian area, which apparently have a broad ecological niche, such as *O. australis* and *P. amazonica*, which were also recorded in the Amazon region of Bolivia, Ecuador and Colombia [26,27]. However, other species, despite not having met the criteria proposed by [25], were also frequent in most of the evaluated localities, such as the species *Akanthomyces tuberculatus*, *Beauveria acridophila*, *Blackwellomyces cardinalis*, *Cordyceps nidus*, *C. tenuipes*, *Ophiocordyceps curculionidae*, *O. sphecocephala* and *Paraisaria gracilis*. Of these, for example, *C. tenuipes* and *P. gracilis* seem to have a wide distribution in neotropical forests cited for Mexico until Bolivia [26,29]. It must be considered that APF follow the distribution patterns of their hosts [31]. Thus, if grasshoppers, spiders, caterpillars and beetles have been dispersed from the Amazon basin to Central America as a result of the isthmus of Panama connecting the two continents, it is reasonable to assume that APF will have a similar distribution. In this regard, we can postulate that with more sampling in the Peruvian Amazon and neighboring countries, we may possibly record them and they may also be considered as hyperdominant as referred to by [32], who postulate tree species such as the palm *Iriartea deltoidea* and *Escheweilera coriacea* are hyperdominant in Amazonian and tropical forests on the mainland, where, incidentally, most entomopathogen populations were recorded in areas where *I. deltoidea* and entomopathogens cohabit.

### 4.3. Influence of Local Environmental Conditions on Arthropod-Pathogen Fungi

According to the relationships with environmental variables at the regional level in the forests of the two regions, elevation was the most significant variable in terms of a good correlation with the richness of APF. This is consistent with the fact that these populations of APF develop in pristine, primary forest habitats and are associated with particular patterns in each microenvironment, as is the case for the genus *Cordyceps* in the foothills of the Colombian Amazon [9,33]. Here, most species of this genus are associated with leaf litter substrates on the ground and on the underside of living leaves close to the ground, where light penetration is very low, as well as being in the understory, even if

there are natural disturbances such as falling trees and insect (ant) infested individuals seek areas with low light penetration. In our case, the elevation gradient in the foothill forests of the Cusco Amazon shows that these microclimates with regimes of temperature, humidity, precipitation and light input to the understory, of which light intensity was the most influential at the local level (Figure 6E). Although it was under the influence of precipitation (r = 0. 23, *p* = 0.16; Figure 6B), this result agrees with other studies where humidity and precipitation are important variables for APF richness as referred to in [11]. In this location, *Cordyceps* and *Beauveria* from the Cordycipitaceae family and *Ophiocordyceps* from the Ophiocordycipitaceae family were the most frequent.

In the family Clavicipitaceae, the genera *Molleriella*, *Nigelia* and *Metharrizium* were found in different kinds of forest. *Nigelia martiale* is more commonly found in undisturbed forests, like the majority samples from Pongo de Qoñec. *Moelleriella phylogena*, known for their ability to parasitize scale insects [34], was common in both pristine and regenerating forests, such as those of samples from Camanti. Finally, the genus *Metarhizium* was poorly represented in anthropized ecosystems, such as the Mazuco locality, in the asexual stage because it is a generalist species. According to [35], in primary and healthy forests, each parasite has well established its host niche, which is too specialized [33]. However, anthropized environments disrupt this balance and allow for the emergence of generalist species such as *Beauveria*, *Metharizium* or *Lecanicillium* that are broadly used in biological control [8], which exhibit a higher rate of reproduction and pathogenicity, but they do not necessarily exhibit a higher lethality rate, as observed in primary forests.

In our case, some samples in the Pongo de Qoñec forest in open areas with intervention, we found very few species with regular numbers of individuals (Figure 6F). In the family Clavicipitaceae, although in the present work we have not dealt with the diversity of arthropod hosts associated with arthropods pathogens, we can corroborate that for *O. australis*, among one of the reasons for its dominance is that this species is almost always hosted by ants known as "isulillas" corresponding to species of *Pachycondyla crassinoda*, which is not only found in primary forest environments but also in open anthropized forests, where these giant ants also obtain their food and adapt to establish their colonies (field observations) [9]. The same pattern occurs in *P. amazonica*, where the host is an orthopteran "brown grasshopper", which also prefers both non-impacted and anthropogenically impacted environments (field observations). This species also appears to have an almost host-specific relationship, which corresponds to groups with a long-standing structured stability. When discussing *P. amazonica*, it is crucial to consider the behavior and life habits of the host. Grasshoppers, belonging to the superfamily Acridioidae, exhibit gregarious behavior only during their immature stages, while as adults, they have a wide range of dispersion and can even cross rivers, i.e., basins [36,37]. Hence, it can be anticipated that the incidence of *P. amazonica* parasitism on grasshoppers would be low. On the contrary, in the case of *O. australis*, whose hosts are hunter ants with social habits, a high abundance of individuals attacked by *Ophiocordyceps* is expected [9]. However, this study reveals a high abundance of this APF, signifying the true dominance of *P. amazonica* and the subjective dominance of *O. australis* associated with the host behavior.

Understanding the distribution of arthropod-pathogenic fungi species in different kinds of forest is essential for developing effective conservation strategies for these fungi, their hosts and the forests they inhabit. Isolating different APF might offer potential for their use in biotechnology for their medicinal properties [5,38] and in agriculture, given their virulence and specificity in pristine forests. However, it should be noted that these fungi may be more sensitive to changes in humidity, temperature and UV light in open agroecosystems [39].

## 5. Conclusions

This first approximation of the knowledge of the high diversity of entomopathogens in southern Peru is still insufficient, but it shows the importance of their conservation and enormous potential for sustainable management.

**Supplementary Materials:** The following supporting information can be downloaded at: https://www.mdpi.com/article/10.3390/d15111122/s1, Table S1: General table of species with collections by basin.

**Author Contributions:** Research conception and design: I.H.-C., M.G.R., T.I.S.G. and M.E.H.R.; field study and sampling: I.H.-C., M.A.Q.O., M.G.R., A.C.M., T.I.S.G., W.Q.A., R.O.P.-A., Z.M.C.C., J.G.M.C. and M.E.H.R.; analysis and interpretation and drafting of the manuscript: I.H.-C., T.I.S.G. and M.E.H.R. All authors have read and agreed to the published version of the manuscript.

**Funding:** This research has been developed thanks to the funding obtained in the scholarship programme: Proyectos Especiales, Formación de Investigadores Postdoctorales en Instituciones Peruanas, with the post-doctoral research entitled "HONGOS ENTOMOPATOGOS NATIVOS POTENCIALES AL CONTROL DE PLAGAS EN SISTEMAS AGROFORESTALES AMAZÓNICOS E IDENTIFICACIÓN DE SUS HÁBITATS VULNERABLES AL CAMBIO CLIMÁTICO EN CUSCO Y MADRE DE DIOS" which was ratified by contract N 061-2021-PROCIENCIA, between Prociencia and the Universidad Nacional de San Antonio Abad del Cusco (UNSAAC).

**Institutional Review Board Statement:** Not applicable.

**Data Availability Statement:** Not applicable.

**Acknowledgments:** The authors give their thanks to CONCYTEC through the PROCIENCIA program; to the CIPHAM laboratory and the vice-rectorate of research of the Universidad Nacional de San Antonio Abad del Cusco; to the Alwyn Gentry Herbarium (HAG), Fundo El Bosque and the Biology Laboratory of the Universidad Nacional Amazónica de Madre de Dios (UNAMAD); to the INKAMAZONIA Ecological Center of the Kosñipata Valley; to the students of the Universidad Nacional Amazónica de Madre de Dios: Jacqueline Lima, Axcel Gutierrez and Jhordi Pumalloclla; to the members of the ECOTAXON Research Center of UNSAAC: Pavel Atauchi, Alex Aite, Abdhiel Bustamante, Anne Arias, Guadalupe Lavi, Nidia Sánchez and Williams Navarro, for their assistance in the field sampling and logistical support; and to the inhabitants of the communities immersed in the Amazonian forests of Cusco and Madre de Dios.

**Conflicts of Interest:** The authors declare that they have no conflict of interest.

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
