# Peer review of "Patterns of Diversity, Structure and Local Ecology of Arthropod-Pathogenic Fungi in the Amazonian Forest of Cusco and Madre de Dios Regions, Southern Peru"

_diversity, doi:10.3390/d15111122_

Round 1
Reviewer 1 Report
This manuscript is well worthy of publication. The amount of sampling is outstanding, particularly for a group as difficult to worth with as entomopathogenic fungi. I have NO concerns regarding the quality of data collection, and the data analyses are excellent and thorough. I went into this knowing which analyses I would do and they did that and then a little bit more. I quite liked all of the figures and find them informative and intuitive. Serious issues with cross-reference between the text and Figure 6 are noted below.
I actually think the authors under-sell the quality and completeness of the research. They focus too much on this being a local phenomenon specific to Peru. While that is indeed the case, I would like to see the authors build on that to develop general principles and things that could be looked for or tested at a more general level (e.g., how could these results impact surveys or interpretations of data of the same fauna in the Afrotropics, Indotropics, or Australotropics)?
For example, in the Introduction: the introduction really focuses in on an objective to characterize this for Peru. But are there more general principles here? Can this be treated as using Peru as a laboratory to understand patterns associated with environmental variables for which hypotheses could be developed to test elsewhere?
And the conclusions: I would like to see a conclusion summarizing what more can be done (e.g. general principles or testable hypotheses) with this and how this paper impacts our overall understanding of environmental and biogeographic correlates of entomopathogenic fungal diversity.
Finally, I have a couple of minor comments:
Lines 197-202 are a little confusing. The authors make it clear that they are reporting the three most species-rich basins in the first sentence. But the second sentence just seems that they are reporting the Chao-1 index, instead of reporting the 3 most diverse ones. I am curious as to why three is the magical number for this paragraph—the top half? Perhaps listing them in order 1-6 would clarify (because low diversity is informative too).
Lines 248-251 need to get the data and figure references correct. Precipitation should be 6b, minimum mean annual temperature should be 6d (R=-0.017, p=0.92), and maximum mean annual temperature should be 6c (R=0.16, p=0.32). Otherwise, this is a very confusing paragraph.
Lines 252-253. Please reference Figure 6e.
Overall, it is quite good. I have made some specific annotations:
Line 34-35. The phrase “elevation is identified as the environmental variable of greatest diversity 34 and structure” is odd. Perhaps “elevation is identified as the environmental variable with the strongest influence on diversity and structure”.
Line 50: perhaps “arthropods including insects” instead of “arthropods and Insects”
Line 58: “found in environments with little impact” to “found in environments with little human disturbance”. The environments don’t have impact, they are impacted.
Line 60: “recorder” to “recorded”
Line 61: “tropical” what? Tropical forests? Tropical latitudes?
Lines 252-257 should be re-punctuated. It reads as though entomopathogen species richness is higher in primary forest habitats and in secondary forests with recent and old anthropogenic impact. This is not what is intended. Rewrite as “this is explained by the fact that entomopathogen species richness is higher in primary forest habitats; in secondary forests with recent and old anthropogenic impact species richness decreases, but a few species are represented by…”
Lines 274-279 is a run-on sentence. Start with a new sentence at “Another genus that was of high…” on line 278.
Line 304 [grammar]: change “there are only two” to “there being only two”.
Author Response
Comments were corrected, all were included in the manuscript.
Reviewer 2 Report
Comments and Suggestions for Authors
This study emphasizes the importance of researching entomopathogens in neotropical forests, especially in the southern Peruvian Amazonian regions of Cusco and Madre de Dios. The study offers a preliminary analysis of the area ecology, entomopathogenic fungal diversity patterns, and entomopathogenic fungi structure. In order to identify the 84 species (including 40% morphospecies), 84 families, 20 genera, and 277 specimens from 39 sites in five basins had to be sampled. This study provides a preliminary investigation of the diversity of entomopathogens in southern Peru. It underlines the need for additional study, the significance of conservation efforts, and the potential for long-term management of these creatures.
The manuscript is well written and presented.
Specific comments:
Introduction: the introduction need to be revised and structured.
The main manuscript contains mentions of all additional remarks. please review the enclosed document.

Minor English language correction is necessary.
Author Response
The observations and suggestions were resolved, they were included throughout the document.
Note: about the reviewer's comment: were molecular data included?
We reply that the analysis of arthropod-pathogenic fungi (APF) for the present research was based on the analysis of morphological and ecological characters, in the future we will work with molecular data, constructing a phylogeny and other data.
Author Response
The observations and suggestions were resolved and included throughout the document.